# “My Health Is More Important than Drinking”: A Qualitative Analysis of Alcohol Use During COVID-19

**DOI:** 10.3390/ijerph22020224

**Published:** 2025-02-05

**Authors:** Cat Munroe, Anthony Surace, Priscilla Martinez

**Affiliations:** 1Department of Psychology, Purchase College SUNY, Purchase, NY 10577, USA; 2Alcohol Research Group, Public Health Institute, Emeryville, CA 94608, USA; asurace@arg.org (A.S.); pmartinez@arg.org (P.M.); 3School of Public Health, University of California, Berkeley, Berkeley, CA 94720, USA

**Keywords:** alcohol use, drinking motives, alcohol use motives, COVID-19, qualitative research

## Abstract

(1) Background: Although alcohol use increased overall in the US during the COVID-19 pandemic, approximately 16% of people decreased their drinking. Understanding reasons for decreasing or discontinuing alcohol use during a time of crisis could inform alcohol messaging during future crises. (2) Methods: We conducted hour-long interviews with 26 participants who reported drinking above NIAAA guidelines at the second wave of the National Alcohol Survey COVID Cohort (a longitudinal study of alcohol use during the COVID-19 pandemic). Data were analyzed using codebook thematic analysis. (3) Results: Many participants reported decreasing use after a period of heavy drinking. Four themes emerged as reasons for doing so: (1) health conditions attributed to or worsened by drinking, (2) concerns about developing the same alcohol problems as a family member, (3) life demands and transitions that limited drinking opportunities, and (4) disliking the side- and after-effects of drinking (e.g., hangovers). (4) Conclusions: Concerns about negative health consequences from heavy alcohol use and limited opportunities to use alcohol due to competing life demands were salient reasons for reducing or abstaining from alcohol use during COVID-19. Incorporating themes about health and life obligations into messaging to reduce alcohol use during crises may improve message relevance.

## 1. Introduction

Research on the global impact of the coronavirus disease 2019 (COVID-19) pandemic on alcohol use shows a polarization of drinking behavior, where some increased their alcohol use and others reduced it [1,2,3,4]. Interestingly, studies from European countries show greater declines in alcohol consumption compared to increases, whereas studies from the United States (US) show greater increases in consumption than decreases [5,6]. Moreover, per capita consumption in the US increased by a historic 3% in 2020 and continued to increase in 2021 by an additional 2.8% [7]. Potential reasons for increased alcohol use in the US during the pandemic are myriad—research has identified economic [8], family [9], and mental health stressors [8] as all impacting alcohol use during this time. Such findings offer further evidence of the influence of stress on substance use (including alcohol) [10]. A substantial body of work has used quantitative methods to identify the characteristics of people for whom alcohol use increased or decreased during the pandemic. US studies have shown any and heavy alcohol use increased among women [11], Black people [6,12], people with mental health struggles [8,13], and people who engaged in heavy drinking before the pandemic [14]. On the other hand, studies have shown decreases in any and heavy alcohol use among young people [3,6] and in any and heavy drinking among men [6], with reductions in opportunities for social engagement due to social distancing measures and business closures and changes in living circumstances as possible drivers of these decreases [15,16]. Identifying those most at risk for deleterious patterns of alcohol use is necessary to reduce the alcohol disparities that may be worsened during times of crisis; however, these data cannot provide insight into the factors that individuals themselves identify as most salient to their decision to reduce their alcohol use or abstain from drinking entirely. Limited qualitative work has investigated the lived experiences and reasons of US residents for limiting their alcohol use during the COVID-19 pandemic as the pandemic progressed. Such studies could provide insight into offering support for decreasing alcohol use or minimizing increases in alcohol use during a global crisis, especially in a national context of overall increasing consumption and well-documented positive associations between stress exposure and alcohol use [10,17].

To date, there have only been a handful of studies examining reasons and motives for reductions in alcohol use during the pandemic in the US, with many from the first year of the pandemic. A quantitative study using a US population-representative sample observed an association between self-reported decrease in or cessation of alcohol use and lower self-reported health-related quality of life (HRQOL) [4]. However, the authors note that the associations between decreases in alcohol use and HRQOL were much less than associations between participants’ reported impact of the pandemic on their lives on HRQOL, suggesting other aspects of the pandemic may be associated with decreases in alcohol use. A longitudinal examination of drinking motives among young people observed that motivations to use alcohol shifted, with increases in drinking to cope with negative emotionality and decreases in drinking for enhancement of positive emotionality; however, the number of weekly drinks consumed did not change [18]. A mixed-methods study among college students showed that shifts in social structures and contexts during the pandemic precipitated reduced drinking, and that use shifted from heavy drinking with peers to lighter drinking with family [16]. Relatedly, young people with high adherence to social distancing guidelines reported using less alcohol compared to young people who did not adhere to such guidelines [19]. While not about reductions in drinking per se, a qualitative study of experiences and perceptions of alcohol and cannabis use among sexual minority women observed that respondents who described increasing their use reported the desire to monitor their alcohol use and identify strategies for reducing their use to avoid developing problem drinking or weight gain [20]. Similarly, a qualitative study of young adult cannabis users in California reported that those who increased either their alcohol or cannabis use did so only temporarily until they adjusted to the “new normal” or adopted more adaptive coping strategies [21].

Taken together, the limited literature in the US suggests that a variety of reasons precipitated reductions in alcohol use among some people during the pandemic. These include changes in social contexts due to social distancing measures (i.e., having to move home from college, reductions in drinking for enhancement, and self-reflection of drinking habits). The existing literature, however, has several limitations. Overall, the research literature is limited in the US relative to other countries, and there is reason to believe the US was unique as one of the few high-income countries to experience an increase in average alcohol use during the COVID-19 pandemic [6,22]. Reasons for pandemic-era reductions in alcohol use may, therefore, be unique to US residents. The extant research has mostly been conducted with community samples or specific populations, limiting generalizability to the general US population. Finally, the vast majority of the research literature on the COVID-19 period is from the first year of the pandemic, which limits understanding of people’s reasons for reducing or limiting alcohol use as the pandemic progressed. An in-depth understanding of individuals’ reasons for reducing alcohol use during the pandemic is necessary to inform future public health strategies to reduce the negative impact of alcohol use during crises. Therefore, we conducted in-depth interviews with select participants of the National Alcohol Survey COVID Cohort (NAS-C19), a 3-wave longitudinal study based on a general population US sample [6]. This study aimed to explore and describe the range of reasons for reducing alcohol use during the COVID-19 pandemic. Our primary research question focused on participants’ reasons for decreasing or discontinuing alcohol use, given their history of heavy drinking during the COVID-19 pandemic. Our analysis focused on both personal reasons for not drinking and shifts in participants’ contexts that were associated with reduced alcohol use.

## 2. Materials and Methods

### 2.1. Sample and Recruitment

The sample consisted of 26 participants who were recruited from the third wave of the NAS-C19, a longitudinal population-based survey of non-institutionalized adults in the US. Data were collected pre-pandemic (2019–2020; Wave 1), mid-pandemic (2021; Wave 2), and after most businesses had re-opened and mask mandates had ended (2022; Wave 3). The initial NAS-C19 sample was collected using probability-based and non-probability-based sampling techniques, and oversampled Black and Hispanic/Latine individuals (see Table 1). A full description of NAS-C19 recruitment methods is available in a study by Kerr et al., 2022 [6]. We asked people who reported past-year drinking in excess of guidelines from the National Institute on Alcohol Abuse and Alcoholism (NIAAA) at Wave 2 after they completed the Wave 3 interview if they would be interested in participating in an hour-long online interview about drinking during the COVID-19 pandemic. Drinking above NIAAA guidelines included reporting 5+ drinks in a single day or >14 drinks per week for men and 4+ drinks in a single day or >7 drinks per week for women. We contacted those who indicated interest by email about participation, and shared information about the interview topic and procedures, logistics such as scheduling the Zoom interview, and the gift card they would receive as an incentive for participation. Participants who did not respond to the invitation email received up to two follow-up emails. All recruitment and interview procedures were approved by the Public Health Institute institutional review board.

### 2.2. Data Collection

Authors 1 and 3 conducted the semi-structured interviews between April and July 2022 via Zoom using a semi-structured interview guide. The interview guide was developed by authors 1 and 3, and focused on participants’ understanding of the relationship between stressful and positive life events and alcohol use in the context of the COVID-19 pandemic. At the start of the appointment, the interviewer described the study rationale and interview topic and obtained verbal informed consent for participation and to record the interview. Interviewers asked participants about their drinking in the past year, aided by a calendar on which they noted major life events in the past year (e.g., family deaths). Interviewers also asked about any stressful experiences (e.g., beginning or leaving a job, moving) that may or may not have been related to COVID-19 (e.g., death in the family, job loss) using 3–5 prompts, depending on the participants’ life events and circumstances. Participants were then asked 11 items focusing on when they were drinking the most and least, what they believed contributed to times they drank more or less than usual, their reasons for drinking or limiting drinking or abstention, and how their alcohol use during the pandemic compared to their alcohol use pre-pandemic. Interviewers also asked participants to describe how the events and circumstances of their lives during the past year were related to their alcohol use, and how their alcohol use affected later life events (if relevant). Interviews lasted approximately 60 min. After the interview, participants were sent an electronic gift card (initially USD 50; USD 75 was offered near the end of the study to increase participation, which the last four participants received). All audio recordings were professionally transcribed for analysis.

### 2.3. Data Analysis

We analyzed transcripts using codebook thematic analysis, which is an iterative technique that allows a broad research question to inform the generation of themes and specifically focuses on centering participants’ experiences and conceptualizations of their experiences, as opposed to interpreting participants’ experiences using an a priori theoretical framework [24]. In this instance, our goal was to understand participants’ experiences of drinking or limiting drinking during the COVID-19 pandemic. Consistent with codebook thematic analysis, authors 1 and 2 independently reviewed and performed open coding of two full transcripts and discussed emerging themes together with author 3. This resulted in a preliminary codebook that authors 1 and 2 used to independently code additional transcripts. As additional transcripts were coded, authors 1 and 2 iteratively added and refined codes to the codebook and discussed emerging themes with author 3. As themes were discussed, if one or more members of the research team identified the need for a new theme, or to separate a theme into two child themes, this was discussed at team meetings, and the resulting decision was noted in the codebook (managed by author 2). This process continued until coding a new transcript did not result in the identification of new themes, at which point saturation was reached and the initial codebook was determined to be complete. After analyzing all transcripts, authors 1 and 2 then reviewed their coding of previously coded transcripts to ensure all codes that were identified in later stages of analysis were applied to all transcripts as appropriate. Then, authors 1 and 2 coded the remaining transcripts. Inter-coder reliability was assessed iteratively—authors 1 and 2 met weekly to compare each coded transcript. Any disagreements in coding were noted and discussed until a consensus was met. This resulted in all transcripts being double-coded and reviewed by all authors, with all coded utterances being the same across transcripts. After, themes were reviewed to determine whether they reached saturation. Codes were discarded if they appeared in a minority of transcripts (e.g., approximately four transcripts or fewer) or did not appear to be specific to the pandemic (e.g., not being “in the mood” to drink).

## 3. Results

The demographics of the sample are reported in Table 1. Overall, participants’ reasons for not drinking fell into four major themes: (1) decreasing their drinking due to concerns that alcohol could negatively affect their physical or mental health; (2) decreasing drinking in order to avoid experiencing the same problems as loved ones who had problems related to alcohol use; (3) being too busy or occupied by other life demands to drink; and (4) decreasing their alcohol use in order to avoid the after-effects of drinking (e.g., hangovers). Please see Appendix A for a summary of themes and exemplar quotes.

### 3.1. Theme 1: Concerns About Alcohol Negatively Affecting Health

Many participants related their decreased drinking to a physical health condition or steps they were taking to address a physical health concern. In some cases, this condition was not causally related to alcohol use but, nevertheless, had an impact on their drinking. For instance, participants described decreasing or ceasing alcohol use due to pregnancy or because they started medication that interacted with alcohol, leading them to decrease their alcohol use. When one participant who described their drinking as decreasing around the time they had surgery during the pandemic was asked how they understood the decrease in her alcohol use, she said, “I probably would say that it was the surgery and recovering from the surgery. Because I think under normal circumstances [my alcohol use] would have been the same” (Participant 1, 44-year-old Hispanic Black woman). Another participant who described her drinking as decreasing and related this to managing a medical illness shared, “Actually, I think I stopped drinking in March because I was on all of these medicines and I was worried about the interactions or the infection” (Participant 22, 25-year-old non-Hispanic White woman).

Although some participants decreased their alcohol use for purely medical reasons and did not have concerns about their drinking causing health problems, others explicitly stated that they cut down on their alcohol use or stopped drinking due to concerns about its impact on their physical health or because their physician stated a medical problem was caused or worsened by drinking. One participant who decreased their alcohol use to address new and severe symptoms of gastroesophageal reflux disease (GERD) shared, “But the GERD is better now. Like, it already improved a lot. But I’m just scared like if I continue drinking the way I used to, I’m going to get sick again” (Participant 25, 33-year-old Hispanic White woman).

Relatedly, a number of participants decreased their alcohol use because they noticed that it worsened other mental health conditions, such as depression or anxiety, or because they were concerned they would develop a substance use disorder. One participant who described alcohol use as related to her depressive symptoms stated the following:
Because like I said, I already suffer from depression, but this just adds, you know, even more to it. It exacerbates all my symptoms. It just, it’s just not helpful at all. But and it’s something I keep thinking about, because it’s like, you know, it’s taking up a lot of time because I’m like, I don’t want to be addicted to alcohol. I don’t want to depend on alcohol.(Participant 10, 40-year-old non-Hispanic Black woman).

These examples demonstrate that participants’ decisions to decrease their alcohol use were often motivated by their desire to preserve or improve their physical and mental health. Although these decisions were not always accompanied by specific concerns about one’s own alcohol use being problematic, it was relatively common for participants to describe alcohol and alcohol use as having potentially disruptive effects on their physical health, sleep, and well-being, and for these concerns to motivate clients to decrease their alcohol use. In addition, a small number of participants shared that when their doctor related problems they were having to their alcohol use, it contributed to their decision to reduce or stop their alcohol use.

### 3.2. Theme 2: Reduce Chance of Experiencing Drinking Problems Faced by Loved Ones

For participants who had a close relationship with someone who had an alcohol use disorder during their lifetime, witnessing the difficulties of their loved one was closely related to their own decision to limit their alcohol use. Participants described this concern both as a reason they did not increase their drinking during times of stress and a motivating factor if they made the choice to reduce their drinking over the course of the pandemic. For example, one participant shared the following:
I’ve seen, through my family, how alcohol has really negatively impacted people. They drink instead of going out to work. They’ll drink in the morning before going to work to kind of help them get their day started and I’ve seen the negative effects of liver damage on family members and things like that as well…and I never wanted to see myself like that.(Participant 7, 36-year-old non-Hispanic Black man).

Another participant describing limiting her use of alcohol described her experience the following way:
I know that there’s other ways to cope... I know that my grandmother was an alcoholic, while my dad was an alcoholic and- and a drug user. But I don’t want to be like them. But it’s the hardest thing.(Participant 10, 40-year-old non-Hispanic Black woman).

In both instances, close relationships with loved ones who had significant problems that were related to alcohol use created motivation for the participant to closely monitor their own use, including keeping a watchful eye on how much and how often they drank to protect their well-being.

### 3.3. Theme 3: Life Demands and Transitions Limiting Drinking Opportunities

Participants attributed decreased drinking or maintaining their usual drinking under circumstances where they would have otherwise expected it to increase, as related to (1) being too occupied by the demands of their life to be able to drink, and (2) maintaining or returning to a consistent routine that did not include drinking. One participant who described being too busy to drink said the following:
…it feels like non-stop busyness. I’m drinking less because there’s so many things going on. And also we’re trying to save money for this wedding because it’s already like getting very expensive. It’s most stressful right now. I’m like the busiest right now, but like it’s not an emotional stressor. I feel very happy. Yeah. But I’m very busy...(Participant 26, 24-year-old non-Hispanic White woman).

Other participants attributed infrequently drinking to maintaining the consistency of their routine from before the pandemic to during the pandemic, which tended not to include drinking, or noted that their alcohol use decreased when they re-established a consistent routine after returning to in-person work or starting a new job as the pandemic progressed. For instance, a participant who described their alcohol use as generally infrequent described their understanding of their use pattern in the following way:
Well, I’m kind of consistent consistently. You know, I have a routine. I maintain a routine. So, it’s like I do that, you know, not going out as much and going out to places just staying home. But staying home hasn’t caused me to drink more like maybe some people that have more time to do that they would drink more. But probably the consistency. I’m pretty consistent in, you know, my daily life, the rituals and that.(Participant 15, 65-year-old Hispanic White man).

Similarly, one participant who returned to working outside of their home after working from home due to the pandemic described this transition and the change in their alcohol use in the following way:
…now I have to leave the house to work. So that prevents me from drinking because when I was stuck here at the house, I would be drinking from the house. Now I really don’t think about it. I don’t really think about that bad habit.(Participant 25, 33-year-old Hispanic White woman).

In each of these examples, responsibilities and commitments were not typically taken on as a strategy to reduce drinking; instead, the structure of the participant’s life created natural barriers to drinking and promoted reductions in drinking overall. It is notable that participants described the structure of their daily lives as creating a guardrail against drinking consistently or frequent episodes of heavy drinking, and that the return to usual, in-person employment once COVID-19 restrictions eased was associated with reduced alcohol use for this reason.

### 3.4. Theme 4: Avoiding Negative Side-Effects and After-Effects of Drinking

Participants described being motivated to reduce their alcohol use during the COVID-19 pandemic due to experiencing myriad negative effects from alcohol use. For example, participants cited hangovers as a deterrent to alcohol use. Participants also attributed feelings of stress or anxiety or problems with memory to their alcohol use. One participant directly tied their decision to not drink for a period to experiencing hangovers, saying, “When I was drinking...liquor, I guess I probably was getting <laughs> hung over a little bit and then I was like, ‘Oh, I’m not going to drink for the week’” (Participant 11, 36-year-old Hispanic White woman). Another participant reflected the following:
If I get drunk every night for four nights in a row, I start to really feel it because I’m getting older. But it starts to drag down my normal day, so I have to quit, I just can’t—if it gets to that point I kind of start feeling it too much and my work starts to suffer and yeah, so it’s always a concern.(Participant 23, 40-year-old non-Hispanic White man).

In both instances, the unpleasantness of hangovers led participants to try to avoid them in the future, either by taking a break from drinking or by reducing the number of days they drank. Although not specific to the context of COVID-19, during the phase of the pandemic where natural barriers to drinking heavily (e.g., working in person, commuting to work) were reduced, the immediate after-effects of drinking heavily and concern about the impact of problems like hangovers may have on work performance also provided some natural barriers to drinking.

## 4. Discussion

Our study explored the reasons behind reductions in alcohol use during the COVID-19 pandemic. Four key themes emerged: concerns about health, avoiding alcohol problems faced by loved ones, life demands limiting drinking opportunities, and avoiding the negative effects of alcohol use. At face value, these reasons do not appear related to the COVID-19 pandemic (e.g., concerns about a family history of substance use disorders could arise at any time). However, we contend it is important to place these themes in the greater context of the pandemic to understand their relevance to reducing drinking during this unique period. Previous study findings show that the pandemic created a context where many of the usual barriers to drinking consistently or heavily were decreased (e.g., physically commuting to/from work and increased alcohol delivery [25,26]). Concurrently, psychological stress increased due to the uncertainty and turbulence of a global pandemic, which led many to use alcohol as a coping strategy and thereby increased their alcohol consumption [11,12]. Given that our findings were derived from interviews conducted in mid-2022 with people who reported drinking in excess of NIAAA guidelines between 2020 and 2021, it is plausible that participants may have begun to experience some of the negative consequences of increased alcohol use, and that this prompted participants to reduce their alcohol use to limit these negative consequences. Thus, while the themes we identified may be common reasons to reduce drinking under any circumstances, it is noteworthy that they became or remained salient as the COVID-19 pandemic progressed. In addition, these findings underscore the significance of increased alcohol use during the first 1–2 years of the pandemic, and the need to prevent heavy alcohol use—and the downstream negative impact on physical and emotional health—during future global emergencies. The multiple contributors to reduced drinking observed by participants and the levels at which they occur converge with ecological systems theory [27], which describes behavior and development as being influenced by multiple contexts, including intrapersonal drives and well-being, interpersonal experiences with loved ones, and larger societal structures (e.g., policies).

Although it remains unclear how pandemic-specific factors influenced the identified reasons for participants’ reduced alcohol consumption, certain contextual factors may have contributed to participants reducing their alcohol use. For example, lockdowns and restrictions in social gatherings may have limited social reinforcement for drinking and facilitated reflection on one’s alcohol use. This idea is supported by research indicating that some people reflected upon their increased alcohol use while quarantining at home [28]. Additionally, a heightened awareness of personal health risks during the COVID-19 pandemic may have also motivated participants’ critical reflection on their alcohol use and to subsequently reduce their alcohol consumption. While these propositions are largely speculative, future research could explore if and how perceived health threats specifically impact alcohol use during times of crises.

These contextual factors can be further understood through the lens of the Health Belief Model [29], which posits that individuals are more likely to change their behaviors if they perceive themselves to be at risk of a health threat (susceptibility), believe the consequences of that threat to be serious (severity), and anticipate that behavior change will offer tangible benefits [29]. While the reality of infection by a new and deadly pathogen took hold during COVID-19, it may also have prompted people to consider the potential health consequences of their alcohol use as more severe than before, and this heightened sense of risk could have prompted them to view reducing alcohol consumption as a beneficial strategy to mitigate those threats. Indeed, our findings—that participants often cited concerns about negative health effects and heightened AUD symptomology—lend some support to this idea. However, it is beyond the scope of these data to demonstrate support for the Health Belief Model. Future research would do well to quantitatively investigate the behavioral mechanisms and pathways through which individual health behaviors may change during times of crises.

Many participants in our study reported that health concerns were a primary motivator for decreasing alcohol use. To date, there is limited research on how alcohol use during the pandemic was impacted by individuals’ health concerns, but there is some research suggesting that social isolation resulting from social distancing facilitated some to reflect on their health behaviors and subsequently reduce their drinking [30,31]. This is somewhat corroborated by research in Australia suggesting an increase in first-time callers to an alcohol helpline [32] and research from the United Kingdom showing increases in the use of “Dry January” apps among those with a potential alcohol use disorder (AUD) [33]. Our findings build on this literature by more directly demonstrating how the pandemic may have precipitated people in the US to consider the health effects of alcohol use, which prompted reduced alcohol use. Our work also shows how these concerns persisted across the later stages of the pandemic and were often tied to the emergence or worsening of existing chronic health conditions (e.g., GERD) or concerns about the impact of alcohol on mental health disorders (e.g., depression and anxiety). Our findings also underscore the salience of physical and emotional health and well-being in the lives of participants and their decision making around alcohol use. The desire to attend to physical and emotional health conditions and responsiveness to physicians’ recommendations to decrease alcohol use were often cited by participants as the key factors that motivated them to decrease their alcohol use. This finding could have significant implications for interventions if reducing health risks does indeed serve as a powerful motivator to decrease alcohol use. These findings highlight the complexity of alcohol use during times of stress, where alcohol may initially be perceived as a coping strategy but can become a concern for long-term health. In this case, the perceived benefit of drinking alcohol may decrease as concerns arise and the negative physical and emotional effects of alcohol use on the body become apparent.

The participants in our study described reducing their alcohol use to avoid the same alcohol problems faced by loved ones. This theme echoes findings from earlier studies that associate a family history of AUD with successfully reducing alcohol use [34]. Participants in our study cited reflecting on the consequences of AUD among family members or friends, leading them to reflect on their own drinking and adopt more mindful drinking habits or abstain altogether. The concerns cited ranged from the personal turmoil loved ones faced related to living with an AUD to witnessing loved ones behave in embarrassing ways while drunk at important events like family gatherings. In order to encourage taking care of oneself while minimizing stigma about individuals with AUDs, framing these concerns as a desire to care for and protect one’s personal well-being is key.

Life demands, particularly being preoccupied with work, family, or other personal responsibilities, also emerged as a key determinant of alcohol use reduction. Participants described how the demands of their daily lives (e.g., managing household responsibilities) left little time for alcohol use. This aligns with previous research indicating that structured routines can serve as protective factors against substance use among those trying to abstain [35]. Conversely, other research may suggest that, for some, alcohol use during the pandemic was facilitated by routines. That is, for some, alcohol use became a habitual inclusion in quotidian responsibilities (e.g., cleaning) during the pandemic, resulting in elevated alcohol use [28]. Taken together, these results suggest that promoting structured activities that require sobriety or are not typically associated with alcohol use could be a potential intervention strategy for reducing alcohol consumption, especially during future crises that entail extended periods of time spent at home (e.g., quarantines or climate emergencies). That is, in-person work may serve as an intervention in and of itself to reduce alcohol use. However, remote work opportunities also increase employment opportunities and accessibility among those with disabilities. Therefore, careful consideration of employment modalities should be made to balance the potential health-promoting benefits of in-person work. Finally, avoiding the negative after-effects of drinking, including hangovers and performance issues at work, was a strong motivator for many participants to reduce their alcohol use. This theme is particularly relevant in the context of the pandemic, where disruptions to work and daily life may have removed some of the natural barriers to heavy drinking (e.g., commuting to work and having more free time) [36]. However, as restrictions lifted, the negative impacts of alcohol use on daily functioning and well-being may have become apparent to many participants, prompting them to scale back their alcohol use. This is somewhat corroborated by research suggesting that alcohol expectancies may have shifted during the pandemic [37], suggesting that the perceived benefits of alcohol use may be outweighed by the negative consequences of its use. For some, these negative consequences may create a natural limit to heavy episodic drinking; however, attention must be paid when these consequences are not associated with decreases in use, as this is consistent with the development of AUD.

### 4.1. Strengths and Limitations

Our study has some notable strengths including the use of a diverse dataset that was pulled from a large population-based survey study. We were able to draw our sample of participants from a larger dataset who reported exceeding NIAAA drinking guidelines in the previous year on a longitudinal survey, which allowed us to recruit a relatively large sample of individuals who drank heavily from a community (vs. clinical) context. This increases our confidence in the utility of our findings, and that the reasons cited for choosing not to drink or reducing drinking are relevant to the population of those who use alcohol above recommended guidelines and not only those who seek treatment for AUDs.

The limitations include the use of participants who completed three waves of a longitudinal survey; although these participants do constitute a general population sample, participants who complete three waves of data collection may differ somewhat from the overall population. The online nature of the interview may also have prevented individuals with limited or inconsistent technology access from participating. Although the research team worked to be very flexible in scheduling the interviews, it is also possible that potential participants who were working outside their homes faced greater barriers to participation, resulting in sampling bias. It is also possible that, in the context of the COVID-19 pandemic, where some participants continued to work from home and may have had privacy limitations during the Zoom interview, some participants did not feel fully able to discuss their alcohol use as a result of privacy concerns, or simply did not feel comfortable participating in a qualitative interview discussing personal matters using a virtual platform. Although participants were generally very forthcoming when discussing their alcohol use, it is also possible that some participants minimized their alcohol use or their reasons for drinking in order to present themselves in a positive light to the interviewer. Many of the participants in the study were highly educated and were not essential workers, allowing them to work from home during the pandemic (see Table 1). As such, it is possible that conclusions drawn from our work are less applicable to those of lower socio-economic status. For instance, the factors associated with decreased alcohol use among those who are drinking heavily despite having the structure of routines that require sobriety may differ. For this reason, future research exploring reasons for decreasing or discontinuing alcohol use among individuals who work out of their homes and are of a lower socio-economic status is key.

### 4.2. Future Directions

Information on individuals’ decisions to reduce or stop their drinking—especially among those who endorsed exceeding NIAAA drinking guidelines for alcohol use in the past year—is particularly important for informing interventions and messaging to support limiting alcohol use. Future research should focus more explicitly on motivations to reduce or eliminate alcohol use, including the physical health concerns and conditions associated with reductions in use (e.g., insomnia), and whether salient motivators to reduce or eliminate alcohol use differ by factors such as race, gender, and sexual identity, as this would allow providers to leverage client-relevant motivators in their recommendations to decrease alcohol use, effectively tailoring their intervention to their clients. Among those whose alcohol use increased during the pandemic and remained elevated for a sustained period of time, it will also be key to understand what factors were and were not effective in supporting a return to pre-pandemic drinking. Relatedly, the finding that care for one’s own physical and mental health and an awareness that alcohol could negatively impact well-being in these areas may have important implications for intervention. Namely, during times of crisis, it may be particularly important to provide information on positive coping strategies alongside information on the long-term physical and emotional effects of heavy alcohol use so that individuals are able to make informed decisions about alcohol use during these high-stress times. Broadening the tools available to engage in emotion-focused coping may be particularly important in the context of a restrictive environment where the situation causing distress cannot be changed (e.g., prolonged social distancing). Similarly, it may be particularly important for public health messaging to include specific information about the negative impact alcohol may have on physical and emotional well-being during times of national and global emergency in order to prevent drinking to cope, which was common in the early phase of the pandemic [11].

## 5. Conclusions

Our qualitative analyses suggest that sampled US-based participants had multiple reasons for reducing their alcohol use during the COVID-19 pandemic. Motivations to reduce alcohol use included participants’ concerns about their health, a family history of alcohol/substance use problems, a lack of time to drink due to their other life demands, and avoidance of the negative side-effects and after-effects of drinking (e.g., hangovers). These findings help to contextualize quantitative research showing changes in alcohol use during the COVID-19 pandemic. Findings suggest that future public health messages aimed at reducing the negative impacts of alcohol during public health crises may do well to emphasize alcohol’s negative impacts on health. Additionally, it may be advisable to emphasize the importance of maintaining a consistent schedule to reduce alcohol use. That is, maintaining a consistent routine may serve as a barrier to excessive alcohol use while in quarantine. Future research is needed to investigate the long-term impacts of pandemic-era changes in alcohol use, and it is vitally important for future investigations to examine how such impacts may manifest differently across different subgroups of the US population.

## Figures and Tables

**Table 1 ijerph-22-00224-t001:** Participant demographics.

Characteristics	Total (*N* = 26) Mean (*SD*) or *n* (%)
Age (range: 23–66)	39.38 (12.7)
Gender	
Female	16 (61.5%)
Male	10 (38.5%)
Race/ethnicity	
Hispanic	8 (30.8%)
White	12 (46.2%)
Black or African American	5 (19.2%)
Other	1 (3.9%)
Marital status	
Married/co-habitating	12 (46.2%)
Divorced	2 (7.7%)
Never married	12 (46.2%)
Education	
High school	1 (3.9%)
Some college	4 (15.4%)
College or more	21 (80.8%)
Income	
USD 70,000 or more	14 (53.9%)
Employed	21 (80.8%)
No alcohol use disorder symptoms *	15 (57.7%)

* Presence of alcohol use disorder symptoms was determined via items derived from the DSM-V [23].

## Data Availability

Data in this manuscript are not readily available due to privacy restrictions related to the personal, sensitive, and potentially identifying information in the qualitative interview transcripts.

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
