# Peer review of "“My Health Is More Important than Drinking”: A Qualitative Analysis of Alcohol Use During COVID-19"

_ijerph, 2025, doi:10.3390/ijerph22020224_

Round 1
Reviewer 1 Report (Previous Reviewer 2)
Comments and Suggestions for Authors
Dear Author
Thank you for making the changes I suggested. Your work is now ready for publication.
Author Response
Reviewer 1
1. Dear Author
Thank you for making the changes I suggested. Your work is now ready for publication.
Response: We are pleased our modifications addressed your concerns. Thank you for your support of our manuscript.
Reviewer 2 Report (Previous Reviewer 1)
Comments and Suggestions for Authors
Firstly, I would like to commend the authors for their detailed revisions and improvements in response to the prior review. It is evident that considerable effort has been made to address the concerns raised previously. The revisions to the title, introduction, and findings sections have significantly enhanced the clarity and academic value of the manuscript. This dedication to refining the study is highly appreciated. That said, there are a few areas where the manuscript can be further strengthened to maximize its impact. Below, I outline five key suggestions for improvement, which I believe can be addressed within a minor revision:
1. Deepening the Discussion of Findings While the findings are organized into four main themes and supported by participant quotations, the underlying psychological, social, and cultural processes driving these motivations require further exploration. Relationships between themes, such as the connection between health concerns and avoiding negative effects, should be analyzed in more detail. Including additional participant quotes to substantiate the themes would enhance the richness and depth of the discussion.
2. Strengthening the Theoretical Framework The findings should be connected to established theoretical frameworks, such as the Health Belief Model or stress and coping theories. Integrating such perspectives would situate the study more firmly within the broader academic discourse. A theoretical foundation would also help explain why and how these motivations to reduce alcohol consumption emerged during the pandemic.
3. Emphasizing the Pandemic Context The manuscript could benefit from a more comprehensive discussion of how pandemic-specific factors (e.g., social isolation, economic uncertainty, and limited access to healthcare) influenced participants’ alcohol consumption. Consideration of how the findings relate to long-term behavioral changes post-pandemic would provide additional depth and relevance.
4. Clarifying Methodological Limitations While Zoom interviews were a practical choice given the pandemic context, their limitations—such as potential restrictions on participants’ comfort or technological access—should be discussed in greater detail. Transparency around the coding process could be improved. For instance, the development of the coding guide and how inter-coder reliability was assessed should be clarified to ensure methodological rigor.
5. Providing Specific Directions for Future Research Suggestions for future research should be more specific. For example, the authors could propose investigating differences in alcohol reduction motivations across demographic groups (e.g., socioeconomic status, age, gender) or examining the long-term effects of the pandemic on alcohol-related behaviors. Developing a clear research agenda that addresses these gaps would strengthen the manuscript’s contribution to the field.
The revisions made by the authors are commendable and have improved the manuscript significantly. The above suggestions focus on enhancing the depth and academic rigor of the paper. These improvements can be addressed through minor revisions, and I am confident that implementing them will further elevate the quality of the manuscript. Once these revisions are completed, I will be happy to recommend the manuscript for publication. I applaud the authors for their diligent efforts and believe this study will make a valuable contribution to the literature.
Author Response
Reviewer 2
Firstly, I would like to commend the authors for their detailed revisions and improvements in response to the prior review. It is evident that considerable effort has been made to address the concerns raised previously. The revisions to the title, introduction, and findings sections have significantly enhanced the clarity and academic value of the manuscript. This dedication to refining the study is highly appreciated. That said, there are a few areas where the manuscript can be further strengthened to maximize its impact. Below, I outline five key suggestions for improvement, which I believe can be addressed within a minor revision:
1. Deepening the Discussion of Findings While the findings are organized into four main themes and supported by participant quotations, the underlying psychological, social, and cultural processes driving these motivations require further exploration. Relationships between themes, such as the connection between health concerns and avoiding negative effects, should be analyzed in more detail. Including additional participant quotes to substantiate the themes would enhance the richness and depth of the discussion.
Response: We thank the reviewer for their feedback. We believe that adding further interpretations of participants’ experiences using an a priori theoretical framework is outside the scope of the thematic analysis framework, and also wish to balance this concern with those of Reviewer 3’s comments regarding centering participant experiences. To address the discrepancies between reviewers’ comments, we expanded the discussion section to include a more in-depth discussion of potential mechanisms through which participants’ alcohol use changed during the pandemic (see lines 282-311). In addition, we have added a supplementary table of additional participant quotes.
2. Strengthening the Theoretical Framework The findings should be connected to established theoretical frameworks, such as the Health Belief Model or stress and coping theories. Integrating such perspectives would situate the study more firmly within the broader academic discourse. A theoretical foundation would also help explain why and how these motivations to reduce alcohol consumption emerged during the pandemic.
Response: We thank reviewer 2 for their considerate feedback. We posit that it is beyond the scope of qualitative data to offer support for behavioral theories/framework. That being said we concur that situating our results within such a model could help contextualize our findings. We have expanded the discussion section to include a passage outlining how the Health Belief Model may help explain our findings (see lines 282-292).
3. Emphasizing the Pandemic Context The manuscript could benefit from a more comprehensive discussion of how pandemic-specific factors (e.g., social isolation, economic uncertainty, and limited access to healthcare) influenced participants’ alcohol consumption. Consideration of how the findings relate to long-term behavioral changes post-pandemic would provide additional depth and relevance.
Response: We thank the reviewer for their feedback. We agree that the consideration of pandemic-specific factors would strengthen our manuscript. We have expanded the discussion section to include an exploration of how COVID-specific contextual factors may have impacted the themes identified throughout the discussion.
4a. Clarifying Methodological Limitations While Zoom interviews were a practical choice given the pandemic context, their limitations—such as potential restrictions on participants’ comfort or technological access—should be discussed in greater detail.
Response: We thank the reviewer for this feedback. We enhanced our description of the potential methodological implications of conducting interviews using the Zoom platform in the following ways: 1) In the “Sample and Recruitment” section, we clarified that the recruitment text invited participants to complete an hour-long online interview (see line 100); 2) In the Discussion section, we further addressed the potential implications of utilizing the Zoom platform, including the potential implications of limiting recruitment to participants who were able to complete the interview online. (see lines 353-362).
4b. Transparency around the coding process could be improved. For instance, the development of the coding guide and how inter-coder reliability was assessed should be clarified to ensure methodological rigor.
Response: We thank the reviewer for their feedback. We have edited the Data Analysis section to provide a step-by-step summary of our processes. This includes more detailed information on our creation of the study codebook and inter-coder reliability determination (see lines 129-145).
5. Providing Specific Directions for Future Research Suggestions for future research should be more specific. For example, the authors could propose investigating differences in alcohol reduction motivations across demographic groups (e.g., socioeconomic status, age, gender) or examining the long-term effects of the pandemic on alcohol-related behaviors. Developing a clear research agenda that addresses these gaps would strengthen the manuscript’s contribution to the field.
Response: We incorporated the reviewer’s recommendations and provided specific directions for future research in the Discussion (see lines 369-387).
The revisions made by the authors are commendable and have improved the manuscript significantly. The above suggestions focus on enhancing the depth and academic rigor of the paper. These improvements can be addressed through minor revisions, and I am confident that implementing them will further elevate the quality of the manuscript. Once these revisions are completed, I will be happy to recommend the manuscript for publication. I applaud the authors for their diligent efforts and believe this study will make a valuable contribution to the literature.
Response: Thank you for your support of our manuscript, and for your time and careful review. We hope to see it published in IJERPH.
Reviewer 3 Report (New Reviewer)
Comments and Suggestions for Authors
The manuscript is both highly relevant and insightful, addressing a relatively understudied topic concerning changes in alcohol consumption behavior during the COVID-19 pandemic, particularly among participants in the United States. I appreciated the clear structure of the study and the way the primary themes were supported by direct participant quotes, which enrich the narrative and provide a nuanced understanding of their experiences. The qualitative approach is a significant strength, capturing details and complexities that might be overlooked in quantitative studies. However, I believe that certain refinements could further enhance the impact and applicability of the findings. Overall, I commend the authors for their valuable contribution.
Key Suggestions
Abstract:
The abstract provides a solid overview but could briefly mention the implications of the findings in public health interventions to attract a broader audience.
Methods:
- Clarify whether the authors created the semi-structured interview or adapted it from a prior source. If it was adapted, cite the original authors. If it was specifically created for this study, provide details about its development.
- Indicate how many questions were included in the interview guide and provide an average duration of the interviews. Regarding the thematic analysis, it would be important to know if there were any codes identified during the process that were ultimately omitted from the analysis. What specific criteria were used to decide whether a code should be included or excluded? Additionally, it would be helpful to detail how the relevance or sufficiency of a new code was evaluated in the context of the study. Providing this information would enhance the analytical process's transparency and strengthen the study's methodological rigor.
Results:
Highlight the importance of centering participants' experiences, which is fundamental in qualitative research. Consider adding a supplementary file with a comprehensive list of participant responses categorized under each code. For example, you could include a table with examples of participant quotes for each code, which would provide a richer understanding of the themes.
Discussion & Limitations:
- Although the text indicates that the identified reasons are not exclusive to the pandemic, it would be helpful to explain in greater detail why these reasons were particularly relevant during the pandemic, beyond the general contextual effects. This could strengthen the connection with the study's findings.
- Critically discuss self-reporting biases and privacy during interviews.
Minor Suggestions
Introduction:
- In line 35, replace “myraid” with “myriad”.
- Line 37, "substance sue" should be corrected to "substance use".
- Line 59, rephrase “decreased or ceased” to “reduced or stopped”.
Materials and Methods:
- Line 152, "noted inthe codebook" should be corrected to "noted in the codebook."
Discussion:
- Use of Abbreviations: Some terms like "AUD" (Alcohol Use Disorder) are not introduced when first used. Add definitions when first mentioned for clarity.
- In line 304, the term "COIVD-19" contains a typographical error and should be corrected to "COVID-19".
Author Response
Reviewer 3
The manuscript is both highly relevant and insightful, addressing a relatively understudied topic concerning changes in alcohol consumption behavior during the COVID-19 pandemic, particularly among participants in the United States. I appreciated the clear structure of the study and the way the primary themes were supported by direct participant quotes, which enrich the narrative and provide a nuanced understanding of their experiences. The qualitative approach is a significant strength, capturing details and complexities that might be overlooked in quantitative studies. However, I believe that certain refinements could further enhance the impact and applicability of the findings. Overall, I commend the authors for their valuable contribution.
Response: Thank you for your thoughtful and careful review of our manuscript. We believe the modifications we made to the paper strengthened it considerably, and hope we have addressed your concerns.
Key Suggestions
Abstract:
1. The abstract provides a solid overview but could briefly mention the implications of the findings in public health interventions to attract a broader audience.
Response: Thank you for your suggestion. Although we would like to extend the abstract, we are unable to do so within the confines of the word limit.
Methods:
2. Clarify whether the authors created the semi-structured interview or adapted it from a prior source. If it was adapted, cite the original authors. If it was specifically created for this study, provide details about its development.
Response: We added additional information focusing on the development of the interview guide to the manuscript in the first paragraph of “Data collection” (see lines 109-111).
3a. Indicate how many questions were included in the interview guide and provide an average duration of the interviews.
Response: We thank the reviewer for this information. This information has now been added to the manuscript in the first paragraph of “Data Collection” (see lines 112-121).
3b. Regarding the thematic analysis, it would be important to know if there were any codes identified during the process that were ultimately omitted from the analysis. What specific criteria were used to decide whether a code should be included or excluded? Additionally, it would be helpful to detail how the relevance or sufficiency of a new code was evaluated in the context of the study. Providing this information would enhance the analytical process's transparency and strengthen the study's methodological rigor.
Response: We thank the reviewer for this observation. Initially reasons for drinking and for not drinking were analyzed; results from the “reasons for drinking” analysis are presented in a separate paper. We added additional information to how codes were selected and removed in the last sentence of the “Data Analysis” section (see lines 143-145).
Results:
4. Highlight the importance of centering participants' experiences, which is fundamental in qualitative research. Consider adding a supplementary file with a comprehensive list of participant responses categorized under each code. For example, you could include a table with examples of participant quotes for each code, which would provide a richer understanding of the themes.
Response: We added the importance of centering participants’ experiences to the first sentence of “Data Analysis,” where thematic analysis is introduced as the analytic framework, so the reader understands that centering participant experiences is a fundamental tenant of our theoretical framework (see lines 125-129). We also added a supplementary table with additional quotes for each theme.
Discussion & Limitations:
5. Although the text indicates that the identified reasons are not exclusive to the pandemic, it would be helpful to explain in greater detail why these reasons were particularly relevant during the pandemic, beyond the general contextual effects. This could strengthen the connection with the study's findings.
Response: We thank the reviewer for this recommendation. We have extended discussion of why participants’ reasons for not drinking were particularly relevant during the pandemic throughout the Discussion.
6. Critically discuss self-reporting biases and privacy during interviews.
Response: Thank you for this suggestion. We integrated discussion of self-reporting biases in lines 360-362.
Minor Suggestions
Introduction:
7. In line 35, replace “myraid” with “myriad”.
Response: This correction was made to the manuscript.
8. Line 37, "substance sue" should be corrected to "substance use".
Response: This correction was made to the manuscript.
9. Line 59, rephrase “decreased or ceased” to “reduced or stopped”.
Response: This correction was made to the manuscript.
Materials and Methods:
10. Line 152, "noted in the codebook" should be corrected to "noted in the codebook."
Response: This correction was made to the manuscript.
Discussion:
11. Use of Abbreviations: Some terms like "AUD" (Alcohol Use Disorder) are not introduced when first used. Add definitions when first mentioned for clarity.
Response: We reviewed the manuscript to ensure all definitions were introduced prior using an abbreviation.
12. In line 304, the term "COIVD-19" contains a typographical error and should be corrected to "COVID-19".
Response: This correction was made to the manuscript.
Round 2
Reviewer 3 Report (New Reviewer)
Comments and Suggestions for Authors
Thank you for carefully addressing the comments provided during the review process. The revisions you have implemented are thorough and effectively address the concerns raised. Pending the editor's final decision, I believe your manuscript meets the necessary standards for publication, and no further changes are required. Congratulations on your work.
This manuscript is a resubmission of an earlier submission. The following is a list of the peer review reports and author responses from that submission.
Round 1
Reviewer 1 Report
Comments and Suggestions for Authors
Thank you for your hard work. I read this article with interest, but I have some concerns:
· The first part of the title ("I Knew I Felt Sick, That I Needed to Stop") is somewhat vague and does not provide a clear indication of what the study is about. Emphasizing the focus on alcohol use and the pandemic earlier in the title could strengthen it. Additionally, the title appears a bit lengthy and complex; expressing it in a more concise and straightforward manner would enhance its impact.
· The introduction does not clearly state the hypothesis or research question. If the research question were explicitly defined in the introduction, the purpose of the study would be more comprehensible.
· The findings are summarized around general themes, but the details within each theme are not thoroughly discussed. In particular, the psychological and social processes underlying the motivations to reduce alcohol consumption could be examined in greater depth. The implications of these themes for understanding the broader meaning and long-term effects of alcohol use should be analyzed in more detail. This would help readers better understand the findings and grasp the significance of the results.
· The generalizability of the findings is limited because the sample consists predominantly of highly educated and socioeconomically advantaged individuals. Considering different demographic groups or socioeconomic conditions would enhance the applicability of the study to a broader population. Given that most participants were able to work from home during the pandemic, the influence of this factor on alcohol use should be discussed more comprehensively.
· The unique conditions of the pandemic (e.g., social isolation, economic uncertainty) and their effects on alcohol consumption are not sufficiently detailed. The relationship between the findings and pandemic-specific factors could be addressed more explicitly.
· The recommendations for future studies are rather general and lack specific guidance. It would be beneficial to clearly indicate which variables or populations should be examined in future research. More detailed suggestions regarding the long-term effects of the pandemic, differences among demographic groups, and strategies for coping with alcohol use could be provided.
· I recommend a more thorough discussion of methodological limitations, such as the challenges associated with conducting interviews via Zoom and potential biases in the sampling methods. Relying solely on Zoom interviews may create an environment where some participants feel more or less comfortable expressing themselves. Supporting the study with alternative data collection methods would provide more comprehensive data.
· More information could be provided about how the coding guide was developed and the criteria used. This would make the thematic analysis process more transparent. Methods for evaluating inter-coder reliability (e.g., statistical measures of agreement) should be specified. Additionally, the discussion could address which themes emerged prominently and which were considered less significant, offering a more detailed account of how the themes were developed.
· In the discussion section, the psychological and social processes underlying the findings could be analyzed more deeply, and connections to the literature could be strengthened. Incorporating a theoretical framework could broaden the interpretation of the findings and provide a better explanation of why and how motivations for reducing alcohol consumption emerged. The impact of pandemic conditions on the study's findings could be discussed in greater depth, along with the unique effects of this period on alcohol consumption.
Author Response
Thank you for your hard work. I read this article with interest, but I have some concerns:
- The first part of the title ("I Knew I Felt Sick, That I Needed to Stop") is somewhat vague and does not provide a clear indication of what the study is about. Emphasizing the focus on alcohol use and the pandemic earlier in the title could strengthen it. Additionally, the title appears a bit lengthy and complex; expressing it in a more concise and straightforward manner would enhance its impact.
Response: We thank the reviewer for this feedback. We have changed the quote in the title to more directly reflect the theme of reducing alcohol use due to health concerns. We have also shortened the title to be more concise.
- The introduction does not clearly state the hypothesis or research question. If the research question were explicitly defined in the introduction, the purpose of the study would be more comprehensible.
Response: We thank the reviewer for this observation. We included an orientation to our research question at the end of the introduction (pgs. 2-3).
- The findings are summarized around general themes, but the details within each theme are not thoroughly discussed. In particular, the psychological and social processes underlying the motivations to reduce alcohol consumption could be examined in greater depth. The implications of these themes for understanding the broader meaning and long-term effects of alcohol use should be analyzed in more detail. This would help readers better understand the findings and grasp the significance of the results.
Response: The implications of our findings for decreased alcohol use and intervention are discussed more extensively in the discussion (pgs. 8-9).
- The generalizability of the findings is limited because the sample consists predominantly of highly educated and socioeconomically advantaged individuals. Considering different demographic groups or socioeconomic conditions would enhance the applicability of the study to a broader population. Given that most participants were able to work from home during the pandemic, the influence of this factor on alcohol use should be discussed more comprehensively.
Response: We underscored the need to identify reasons for decreasing/discontinuing alcohol use among individuals who have the structure of out-of-home work and are of a lower socio-economic status (pg. 9).
- The unique conditions of the pandemic (e.g., social isolation, economic uncertainty) and their effects on alcohol consumption are not sufficiently detailed. The relationship between the findings and pandemic-specific factors could be addressed more explicitly.
Response: Thank you for this observation. Throughout the discussion in particular, we worked to show how the context of participants’ lives shifted over time, and how contexts such as lockdown and working from home vs. Re-initiating usual routines affected alcohol use. In most cases, participants themselves did not explicitly draw parallels between the changes in their larger contexts and their alcohol use, which limits the extent to which our analyses can argue for these associations based on the thematic analysis framework. We hope that we have sufficiently contextualized participants’ responses in the COVID-19 context to address the reviewer’s concerns.
- The recommendations for future studies are rather general and lack specific guidance. It would be beneficial to clearly indicate which variables or populations should be examined in future research. More detailed suggestions regarding the long-term effects of the pandemic, differences among demographic groups, and strategies for coping with alcohol use could be provided.
Response: We noted more specifically the need to identify physical health concerns associated with reducing/ceasing alcohol use, as well as more focused discussion of preventing drinking to cope.
- The recommendations for future studies are rather general and lack specific guidance. It would be beneficial to clearly indicate which variables or populations should be examined in future research. More detailed suggestions regarding the long-term effects of the pandemic, differences among demographic groups, and strategies for coping with alcohol use could be provided.
Response: We noted more specifically the need to identify physical health concerns associated with reducing/ceasing alcohol use, as well as more focused discussion of preventing drinking to cope (see page 9).
- I recommend a more thorough discussion of methodological limitations, such as the challenges associated with conducting interviews via Zoom and potential biases in the sampling methods. Relying solely on Zoom interviews may create an environment where some participants feel more or less comfortable expressing themselves. Supporting the study with alternative data collection methods would provide more comprehensive data.
Response: We expanded the discussion of potential privacy limitations and sampling bias in the “strengths and limitations” section of the discussion (pg. 8).
- More information could be provided about how the coding guide was developed and the criteria used. This would make the thematic analysis process more transparent. Methods for evaluating inter-coder reliability (e.g., statistical measures of agreement) should be specified. Additionally, the discussion could address which themes emerged prominently and which were considered less significant, offering a more detailed account of how the themes were developed.
Response: We discussed the process of developing the codebook and the iterative process of coding (pgs. 3-4). Because each transcript was reviewed by both coders and any discrepancies in coding were discussed and resolved between coders, there was certainty regarding consensus by the end of the coding process. Generally, themes that reached saturation were presented in the paper; those that did not reach saturation were not included.
- In the discussion section, the psychological and social processes underlying the findings could be analyzed more deeply, and connections to the literature could be strengthened. Incorporating a theoretical framework could broaden the interpretation of the findings and provide a better explanation of why and how motivations for reducing alcohol consumption emerged. The impact of pandemic conditions on the study's findings could be discussed in greater depth, along with the unique effects of this period on alcohol consumption.
Response: We thank the reviewer for this observation. Because thematic analysis explicitly avoids analyzing data from within an existing theoretical framework, applying specific frameworks to psychological and social processes described by participants is out of the scope of the current project (See Braun & Clarke’s 2006 article, “Using Thematic Analysis in Psychology,” published in Qualitative Research in Psychology). Similarly, hypothesizing why participants’ concerns emerged, beyond their own descriptions and rationales they provided in the interview, is outside the guidelines of our analytic framework. However, we do wish to address the reviewers’ concern; therefore, we integrated brief discussion of Bronfenbrenner’s ecological systems theory into the discussion in order to situate participants’ behaviors within the larger context of their lives, including the COVID-19 period (pg. 7).
Reviewer 2 Report
Comments and Suggestions for Authors
The study is well organised and presents an essential topic on reducing alcohol consumption during the COVID-19 pandemic. I must say that I found the topic exciting and that reading the article was remarkably smooth and enjoyable. I believe the manuscript can be considered for publication once the minor changes suggested below have been made.
Abstract
Remove numbers in brackets before “Background, Methods, Results and Conclusion”.
Introduction
The authors should briefly introduce the influence of the COVID-19 pandemic in the genesis and/or exacerbation of stress-related diseases. Stress itself can influence the consumption of different psychotropic substances, including alcohol.
https://doi.org/10.1016/j.janxdis.2020.102232
https://doi.org/10.1007/s002130100917
https://doi.org/10.1080/07853890410018862
https://doi.org/10.26719/emhj.22.024.
Line 78-85: The authors might consider making this part more homogeneous. Suggest avoiding using an ordered or sequential list.
2.1 Sample and Recruitment
The methodology is clear, but it might help to highlight more explicitly the demographic diversity of the sample earlier. This can help contextualise the study's external validity.
Data collection, data analysis and results are good enough.
Discussion
Line 303 – 307: Authors could discuss in more depth how health concerns (risk of liver disease and/or mental disorders) may have influenced the drinking patterns.
Line 319 – 324: The discussion on reducing alcohol consumption to avoid dealing with the alcohol problems faced by loved ones is brief and could be addressed more comprehensively.
Author Response
- The study is well organised and presents an essential topic on reducing alcohol consumption during the COVID-19 pandemic. I must say that I found the topic exciting and that reading the article was remarkably smooth and enjoyable. I believe the manuscript can be considered for publication once the minor changes suggested below have been made.
Response: Thank you for your kind feedback on our manuscript.
Abstract
- Remove numbers in brackets before “Background, Methods, Results and Conclusion”.
Response: Our understanding of IJERPH’s formatting guidelines is that the numbers in brackets must remain. If this is not the case, we will happily remove them.
Introduction
- The authors should briefly introduce the influence of the COVID-19 pandemic in the genesis and/or exacerbation of stress-related diseases. Stress itself can influence the consumption of different psychotropic substances, including alcohol.
- https://doi.org/10.1016/j.janxdis.2020.102232
- https://doi.org/10.1007/s002130100917
- https://doi.org/10.1080/07853890410018862
- https://doi.org/10.26719/emhj.22.024.
Response: We agree that stress related to the COVID-19 pandemic is likely related to
documented increases in alcohol use. In order to balance the need to thoroughly review the extant literature and maintain the focus on reductions in alcohol use during COVID-19, we added two of the proposed citations to the introduction and make a clear note of the
association between stress and alcohol use, as relevant to COVID-19 (pg. 2).
- Line 78-85: The authors might consider making this part more homogeneous. Suggest avoiding using an ordered or sequential list.
Response: We updated this paragraph to avoid the use of an ordered/sequential list and also edited the writing so it is in a style that is more consistent with the rest of the manuscript.
2.1 Sample and Recruitment
- The methodology is clear, but it might help to highlight more explicitly the demographic diversity of the sample earlier. This can help contextualise the study's external validity.
Response: To better orient the reader to the demographic diversity of the sample earlier in the paper, we added a reference to Table 1 in the second sentence of the Methods section (see page 3).
Data collection, data analysis and results are good enough.
Response: We thank the reviewer for their positive feedback.
Discussion
- Line 303 – 307: Authors could discuss in more depth how health concerns (risk of liver disease and/or mental disorders) may have influenced the drinking patterns.
Response: we expanded our discussion of health concerns and drinking in the discussion (pg. 7).
- Line 319 – 324: The discussion on reducing alcohol consumption to avoid dealing with the alcohol problems faced by loved ones is brief and could be addressed more comprehensively.
Response: We also extended our discussion of alcohol problems faced by loved ones (pg. 7).